# Health and well-being needs of Indigenous adolescents: a protocol for a scoping review of qualitative studies

Andrew Sise [1], Peter Azzopardi [2,3] Alex Brown,[4,5] Jordan Tewhaiti-Smith [6]
Seth Westhead,[7] Jaameeta Kurji,[7,8] Daniel McDonough,[7] Rachel Reilly [7,9]
Brittany Bingham,[10,11] Ngiare Brown,[7] Chenoa Cassidy-Matthews,[12]
Terryann C Clark [13] Salenna Elliott,[7] Summer May Finlay,[14]
Ketil Lenert Hansen,[15] Matire Harwood [16] Jonill Margrethe Fjellheim Knapp,[17]
Siv Kvernmo,[18] Crystal Lee,[19] Ricky-Lee Watts,[20] Melanie Nadeau,[21]
Odette Pearson [7] Jeff Reading,[22] Elizabeth Saewyc [23] Amalie Seljenes,[17]
Jon Petter A Stoor [24,25] Paula Aubrey,[26] Sue Crengle [1]

For numbered affiliations see end of article.

**Correspondence to**
Prof. Sue Crengle;
sue.crengle@otago.ac.nz

## ABSTRACT

**Introduction** Improving the health of Indigenous adolescents is central to addressing the health inequities faced by Indigenous peoples. To achieve this, it is critical to understand what is needed from the perspectives of Indigenous adolescents themselves. There have been many qualitative studies that capture the perspectives of Indigenous young people, but synthesis of these has been limited to date.

**Methods and analysis** This scoping review seeks to understand the specific health needs and priorities of Indigenous adolescents aged 10–24 years captured via qualitative studies conducted across Australia, Aotearoa New Zealand, Canada, the USA, Greenland and Sami populations (Norway and Sweden). A team of Indigenous and non-Indigenous researchers from these nations will systematically search PubMed (including the MEDLINE, PubMed Central and Bookshelf databases), CINAHL, Embase, Scopus, the Informit Indigenous and Health Collections, Google Scholar, Arctic Health, the Circumpolar Health Bibliographic Database, Native Health Database, iPortal and NZresearch.org, as well as specific websites and clearinghouses within each nation for qualitative studies. We will limit our search to articles published in any language during the preceding 5 years given that needs may have changed significantly over time. Two independent reviewers will identify relevant articles using a two-step process, with disagreements resolved by a third reviewer and the wider research group. Data will then be extracted from included articles using a standardised form, with descriptive synthesis focussing on key needs and priorities. This scoping review will be conducted and reported according to the Preferred Reporting Items for Systematic Reviews and Meta-Analyses extension for Scoping Reviews guidelines.

**Ethics and dissemination** Ethics approval was not required for this review. Findings will be disseminated via a peer-reviewed journal article and will inform a broader international collaboration for Indigenous adolescent health to develop evidence-based actions and solutions.

## STRENGTHS AND LIMITATIONS OF THIS STUDY

⇒ The planned scoping review will explore qualitative literature that identifies the perspectives of Indigenous adolescents.

⇒ International literature will be synthesised to identify differences and commonalities in needs between Indigenous youth living in different regions and contexts.

⇒ The Global Collective undertaking this work is led by Indigenous people, and includes Indigenous and non-Indigenous researchers and youth advocates from across included nations.

⇒ A limitation of this review is the potentially limited evidence for younger adolescents and for some included Indigenous peoples.

⇒ A further limitation is that this review is restricted to Indigenous peoples in included high-income countries.

## BACKGROUND

Indigenous peoples, numbering nearly 500 million individuals worldwide,[1] inhabit diverse regions and possess a rich variety of cultures, languages, and traditions.[2]

Indigenous peoples have the right to enjoy the highest attainable standards of physical and mental health.[3] Yet globally, they experience health outcomes that are substantially poorer than those of non-Indigenous populations.[4] These differences, termed inequities due to their being both avoidable and unjust,[5] are manifestations of a broad range of challenges that have been faced by Indigenous peoples. Central to the development of these inequities has been the process of colonisation.[6] Since the first arrival of settlers, Indigenous peoples have been displaced from their lands, and experienced the destruction and

degradation of natural environments. New infectious diseases and harmful substances, such as alcohol and tobacco, have been introduced often with devastating consequences. Long-established traditions and practices have been suppressed and disrupted by unjust laws and regulations, and Indigenous peoples have been marginalised within dominant settler societies. Colonisation, harmful policies and racism are not confined to the past, continuing into the present day.[7] This has had, and continues to have, a profound impact on the health of Indigenous populations.[6]

Despite this Indigenous peoples have resisted, survived and maintained their distinct identities as sovereign peoples.[2] Indigenous peoples also continue to hold unique knowledge and worldviews. Embracing and respecting these ways of knowing alongside Western paradigms of health and knowledge generation[8] will provide substantial opportunities for Indigenous health advancement. Although it is imperative that governments and institutions act to improve the health of Indigenous peoples, it is essential that Indigenous voices and perspectives are at the forefront of these efforts.[2]

In order to address inequities faced by Indigenous peoples, it will also be crucial to consider the health and well-being of Indigenous adolescents, who now represent one-third of most Indigenous populations.[9] While adolescence has traditionally been considered the healthiest time in a person's life, we now understand it as a life phase where unique health needs emerge, particularly those related to sexual and reproductive health, mental health, substance use and injury.[10] Adolescence is a pivotal time in physical, social and emotional development, and a time when individuals develop their identity; this results in adolescents being particularly susceptible to racism and discrimination based on ethnicity, religion, disability, sexual orientation, gender expression or other factors. It is a time when health risk behaviours prominently emerge, and are potentially modifiable. Health and well-being during adolescence is an important determinant of intergenerational health.[11] Crucially, adolescents are also incredible agents of change, and bring unique skills and perspectives that can shape health and well-being for all.[12] As such, adolescence is increasingly recognised as a time at which investments in health and well-being can bring maximal benefits, both now and into the future.[10]

Despite its clear importance, global investment in adolescent health to date has been limited, at least in part due to a lack of data regarding adolescent health needs.[10] Currently, efforts such as the WHO's Global Action for Measurement of Adolescent Health[13] are underway, aiming to strengthen quantitative measurement of adolescent health needs. The success of these efforts, however, will remain dependent on the quality of data available at the individual country level. In addition, Indigenous adolescents experience different health risks and experience substantial health inequities when compared with non-Indigenous adolescents, which may not be captured by existing data and indicator systems.[6 9] To achieve the crucial task of improving health outcomes for Indigenous adolescent people, a specific focus on their health and well-being, which recognises their unique status as Indigenous peoples will also be required.

Presently, the Global Collective for Indigenous Adolescent Health and Evidence-Based Action (the Global Collective) is undertaking a project that seeks to enhance knowledge in this area and advocate for evidence-based action to improve Indigenous adolescent health.[12] This will involve both identifying relevant indicators through which Indigenous adolescent health can be monitored, and assembling the best available evidence to develop actions with which to respond. Initially, the Global Collective will focus on Indigenous adolescents (defined for this purpose as being 10–24 years of age) living within Australia, Aotearoa New Zealand, Alaska, Canada, the USA, Greenland, Norway and Sweden. The Global Collective is led by Indigenous people, and includes Indigenous and non-Indigenous researchers and youth advocates from across these nations.

As a part of this effort to measure and respond to Indigenous adolescent health, it is crucial to understand what is important for health and well-being from the perspectives of Indigenous adolescents themselves. Qualitative research is an ideal format for ascertainment of such information, because it allows researchers to hear from young people in their own voice and will enable young people to describe their own priorities, needs and solutions. A preliminary search has identified that there is an appreciable body of qualitative literature that is focused on Indigenous adolescent health and well-being; this is also consistent with an earlier mapping of data available for Indigenous adolescents in Australia.[14] For example, qualitative studies have explored the health needs of Aboriginal and Torres Strait Islander adolescents living in rural settings in Australia,[15] perspectives of First Nations adolescent girls in Canada regarding the meaning of (and what is important for) a healthy body[16] and Inuit youth perspectives on sexual health in Nunavut.[17] Recently, substantial progress has been made in the synthesis of this evidence, with a systematic review of peer-reviewed literature having examined and compared key aspects of well-being for Indigenous young people aged 18 years or less in Canada, Australia, Aotearoa New Zealand and the USA.[18] To build on these efforts and set the foundation for the planned work of the Global Collective,[12] further exploration is necessary. In particular, it will be important to identify and map evidence across all nations included within the Global Collective's workplan, identify literature for youth across the Global Collective's broader age definition of adolescence and explore the appreciable body of grey literature that may be available for some nations. Scoping reviews are well suited to exploring and identifying key concepts within a body of literature, as well as mapping, reporting and discussing these concepts.[19]

This protocol is for a scoping review that aims to map the available qualitative literature for Indigenous adolescents in Australia, Aotearoa New Zealand, Canada, the

USA, Greenland, Norway and Sweden. We seek to map qualitative data that describe key issues and needs from the perspective of adolescents themselves.

## METHODS

The development of this protocol was informed by the Joanna Briggs Institute Scoping Review Methodology Guidelines.[20] This scoping review will be reported in accordance with the Preferred Reporting Items for Systematic Reviews and Meta-Analyses extension for Scoping Reviews (PRISMA-ScR) checklist.[21] The scoping review that this protocol describes is underway. We ran the preliminary search in electronic databases in October 2023 and to date have completed a title and abstract screen based on eligibility criteria. Full texts are currently being retrieved for further assessment and we will also hand search grey literature. The review is being undertaken by researchers across the Global Collective, with work relevant to each nation being undertaken by researchers and experts of that nation. We anticipate the review will be completed in late 2024.

### Eligibility criteria
#### Participants
Eligible studies will include participants who are Indigenous peoples of Aotearoa New Zealand, Alaska, Australia, Canada, the USA, Greenland, Norway, or Sweden, and are between 10 and 24 years of age.

As per United Nations practice,[2] a rigid definition of Indigenous status will not be used. Participants of a study will be considered Indigenous when this is clearly identified by the text of the associated article. While it is acknowledged that Indigenous peoples inhabit many regions of the world, the included regions/nations of Aotearoa New Zealand, Alaska, Australia, Canada, the USA, Greenland, Norway and Sweden have been chosen for the initial work of the Global Collective due to similarities in their colonial histories, data systems and policy contexts.[12] Currently, the WHO defines adolescence as occurring between the ages of 10 and 19 years.[22] However, evolving understandings of the timing of biological maturation, and increasing delays in role transitions such as marriage and parenthood mean that it is now proposed that adolescence should be defined as occurring between the ages of 10 and 24 years.[23] The Global Collective has chosen the age range of 10–24 years to reflect this understanding, and to capture the profound health-shaping changes which occur during this pivotal stage of development.[12] For this review, we will only include studies that specifically sample and report data for individuals ranging from 10 to 24 years of age.

### Concept
To be eligible, studies must report qualitative data from participants' perspectives on their health and well-being.

### Context
To be eligible, a study must have been conducted within Aotearoa New Zealand, Australia, Canada, the USA, Greenland, Norway, or Sweden. Any context or setting within these nations or regions will be acceptable.

### Publication type
We will include peer-reviewed and grey literature; in Australia, for example, there are a number of high-quality consultations with Aboriginal and Torres Strait Islander adolescents that are published by Government and non-Government organisations.[24] To be eligible, studies must report data attained using qualitative methods (eg, interviews, focus groups, photovoice) where youth participants were able to identify and express their own views and perspectives. Quantitative studies including, for example, analyses of responses to surveys that used pre-set questions determined by researchers will not be included. Mixed methods studies (including both qualitative and quantitative components) will be eligible only if the health needs and/or priorities that are presented were identified based on qualitative enquiry with Indigenous adolescent people.

### Date and language limitations
Publications in any language, published during the five calendar years prior to, and the year up until the search date will be eligible for inclusion. Evolving society, technology and culture means that the determinants of adolescent health and well-being are likely to be changing over time. The window for study inclusion has been chosen to reflect this, as more contemporary studies will likely be of greater relevance. If insufficient articles are identified with publication dates within this period, this window may be extended to include earlier years.

### Search strategy
This scoping review will follow a three-part search strategy. First, databases (including Google Scholar) will be systematically searched for relevant literature. Second, key journals, websites and clearinghouses relevant to Indigenous health in each Nation will be hand searched. Third and finally, the references of all included articles will be searched to identify any additional relevant articles that were not retrieved by the first two steps. Our search is also to be completed by a network of experts across each Nation who will bring knowledge of available resources and publications.

### Databases
A systematic electronic search will be performed of PubMed (including the MEDLINE, PubMed Central and Bookshelf databases), CINAHL, Embase, Scopus, Informit (Indigenous and Health collections), Google Scholar, Arctic Health, the Circumpolar Health Bibliographic Database, Native Health Database (Canada and USA), iPortal (University of Saskatchewan) and NZresearch.org (Aotearoa New Zealand).

### Search terms
The full search terms which will be applied to PubMed, CINAHL, Embase, Scopus and the Informit Indigenous

and Health collections are provided in online supplemental file 1. These search terms were primarily developed in the PubMed search engine using an iterative process consisting of repeated searches, analysis of included and excluded titles and analysis of terms used in article keywords, titles and abstracts. Relevant Medical Subject Heading (MeSH) terms were identified using the National Library of Medicine's MeSH browser,[25] and search terms for identifying qualitative literature were adapted from terms presented in a 2017 analysis by Rogers *et al.*[26] Simple keyword (non-controlled-vocabulary) terms were translated from the PubMed search string to all other databases directly. For databases that used controlled vocabularies different from MeSH terms (CINAHL and Embase), searching of those vocabularies to identify equivalent or additional relevant vocabulary terms was also completed.

### Selection of sources of evidence to include in the review

Once the searches have been conducted in all databases and duplicate citations removed, the remaining citations will be screened using Covidence systematic review management software. Initially, titles and abstracts will be screened for potential relevance. The full texts of potentially relevant articles identified at the title/abstract screening stage will then be retrieved for assessment in detail against the inclusion criteria of the review. At both the title/abstract and full-text stages, two independent reviewers will be used, with any disagreements resolved by a third reviewer. In the final reporting of the review, a flow diagram will be used to document the selection of articles for inclusion, and reasons for the exclusion of articles at the full-text stage will be documented in accordance with the PRISMA-ScR guidelines.[21]

### Hand searching of specific websites, clearinghouses, journals and references

Once the electronic search is complete, additional hand searching of specific websites, clearinghouses and journals relevant to Indigenous health, and the references of all studies identified as being eligible for the review, will be completed. Journals to be hand searched will include: *AlterNative*, the *Journal of Indigenous Wellbeing: Te Mauri Pimatisiwin, MAI Journal, ab-Original: Journal of Indigenous Studies and First Nations and First Peoples' Cultures,* the *International Journal of Indigenous Health, American Indian and Alaska Native Mental Health Research,* The *American Indian Culture and Research Journal,* the *Canadian Journal of Native Studies,* and the *International Journal of Circumpolar Health.*

### Data extraction

Data extraction will be handled in two stages. First, for studies meeting inclusion criteria (Indigenous population AND adolescent focused AND qualitative in design) we will capture the basic demographics of the study using a standardised form. This will include publication year, abbreviated reference and whether the study focuses on understanding health and well-being, needs

and priorities from the perspective of adolescents OR if the study focuses on a specific health issue/s (such as mental health). While this review is restricted to studies in which youth were able to identify and express their own views and perspectives, we will also consider the extent to which studies appear to reflect these perspectives and whether, for example, core findings appear based primarily on youth or the researcher's recommendations. In the second phase, we will first look at those studies that explore broad issues, needs and priorities, such as that conducted by Mohajer and colleagues.[15] For these studies, we will extract the aims and research questions, qualitative method(s), participants (including number, Indigenous people(s), age range, gender(s) represented), setting (including country/region) and the quality of the study from an Indigenous research perspective.[27] For issue-specific qualitative research, we will extract similar information but also map the specific issues that they focus on; we will use the priority needs framework for adolescents globally as developed by WHO as a guide for this mapping.[28] The relative focus of qualitative studies against this framework will also provide some insight into relative needs and priorities.

### Analysis of evidence and presentation of results

Analysis and presentation of results will be primarily descriptive, and efforts to assess the level of certainty of results or undertake more complex methods of synthesis such as meta-aggregation (as might be done in a qualitative systematic review) will not be undertaken.[20]

First, we will map the nature of the qualitative data overall for Indigenous adolescents. A descriptive table will be used to describe key characteristics of included studies including time of publication, age of participants, Indigenous peoples covered and the focus of these qualitative studies. This will enable the available literature to be mapped and gaps in coverage to be identified.

We will then undertake further analyses of the studies identified by this review. Due to the wide scope of included nations and data sources, these analyses will be inherently exploratory, and their final format will depend on the quality and nature of the data identified. However, we hope to be able to synthesise the overall priorities and needs of Indigenous adolescents that have been identified from across the findings of our included studies. We plan to use a simple descriptive analysis of qualitative content[20] to synthesise the health and well-being needs and priorities that are identified from the findings of the included literature. We will aim to code needs and priorities identified by individual studies (eg, stress, worry, poor sexual health, racism), then group these into higher-level categories (eg, health outcome, health risk, health determinant), using a process similar to that undertaken by Harfield and colleagues.[29] Alongside describing any core health and well-being needs and priorities that are apparent across the findings of many studies, these categories will enable (where data are sufficient) simple comparisons across different demographic variables such

as age group, country/region and gender. A narrative description will be used to highlight the key findings of these analyses.

## Patient and public involvement

There was no patient or public involvement in the development of this protocol.

## Ethics and dissemination

Ethical approval is not required for this scoping review. Dissemination will occur via two key avenues. First, a report of the review and its findings will be published as a standalone peer-reviewed journal article. Second, findings from this review will inform work currently underway by our Global Collective to identify key indicators for monitoring Indigenous adolescent health across nations, and synthesise evidence to develop appropriate responses.[12] The findings and recommendations of this wider project are intended to be published in an upcoming two-part series of journal articles.

## CONCLUSION

Improving the health and well-being of Indigenous adolescents will be essential to addressing the inequities faced by Indigenous peoples.[9] With this scoping review, we will evaluate the peer-reviewed qualitative literature pertaining to Indigenous adolescent perspectives on what is important for their health and well-being. Having begun with a preliminary search in October 2023, this review is progressing with work relevant to each nation being led by researchers and experts of those nations. This will enable consideration of issues specific to each region, while also allowing the identification of common themes and the development of overarching recommendations. By identifying what is important to adolescents themselves, this review will highlight areas for action. It will also map the coverage of literature in this crucial area, and establish whether gaps exist that warrant further research. Depending on the review's findings this could include, for example, identifying priority areas for consultation and primary qualitative enquiry, or setting the direction for further evidence syntheses to develop evidence-based responses. This will be a key step in the wider efforts of the Global Collective,[12] which aims to measure, respond to and develop strategies to enhance Indigenous adolescent health.

## Author affiliations

[1]Ngāi Tahu Māori Health Research Unit, University of Otago, Dunedin, Aotearoa New Zealand
[2]Adolescent Health and Wellbeing, Telethon Kids Institute, Adelaide, South Australia, Australia
[3]Centre for Adolescent Health, Murdoch Children's Research Institute and Department of Paediatrics, The University of Melbourne, Melbourne, Victoria, Australia
[4]National Centre for Indigenous Genomics, Australian National University, Canberra, Australian Capital Territory, Australia
[5]Telethon Kids Institute, Nedlands, Western Australia, Australia
[6]Medical Research Institute of New Zealand, Wellington, Aotearoa New Zealand
[7]Wardliparingga Aboriginal Health Equity Theme, South Australian Health and Medical Research Institute, Adelaide, South Australia, Australia
[8]School of Epidemiology & Public Health, Faculty of Medicine, University of Ottawa, Ottawa, Ontario, Canada
[9]School of Psychology, The University of Adelaide, Adelaide, South Australia, Australia
[10]Faculty of Medicine, Division of Social Medicine, University of British Columbia, Vancouver, British Columbia, Canada
[11]Centre for Gender & Sexual Health Equity, University of British Columbia, Vancouver, British Columbia, Canada
[12]School of Population and Public Health, The University of British Columbia, Vancouver, British Columbia, Canada
[13]School of Nursing, Faculty of Medical and Health Sciences, University of Auckland, Auckland, Aotearoa New Zealand
[14]School of Health and Society, University of Wollongong, Wollongong, New South Wales, Australia
[15]Regional Centre for Child, Youth Mental Health and Child Welfare North (RKBU North), Faculty of Health Sciences, UiT - The Arctic University of Norway, Tromsø, Norway
[16]Department of General Practice and Primary Care, University of Auckland, Auckland, Aotearoa New Zealand
[17]Faculty of Health Sciences, UiT - The Arctic University of Norway, Tromsø, Norway
[18]Department of Clinical Medicine, UiT - The Arctic University of Norway, Tromsø, Norway
[19]College of Population Health, University of New Mexico, Albuquerque, New Mexico, USA
[20]None, Port Alberni, British Columbia, Canada
[21]Department of Indigenous Health, School of Medicine and Health Sciences, University of North Dakota, Grand Forks, North Dakota, USA
[22]Faculty of Health Sciences, Simon Fraser University, Vancouver, British Columbia, Canada
[23]School of Nursing, University of British Columbia, Vancouver, British Columbia, Canada
[24]Department of Epidemiology and Global Health, Lávvuo-Research and Education for Sámi Health, Umeå University, Umeå, Sweden
[25]Centre for Sami Health Research, Department of Community Medicine, UiT - The Arctic University of Norway, Tromsø, Norway
[26]Indigenous Health Department, School of Medicine and Health Sciences, University of North Dakota, Grand Forks, North Dakota, USA

**Acknowledgements** The authors would like to thank Christy Ballard, Librarian at the University of Otago, for her input and review of the search strategy for this protocol.

**Contributors** All authors (ASi, PAz, AB, JT-S, SW, JK, DM, RR, BB, NB, CC-M, TCC, SE, SMF, KLH, MH, JMFK, SK, CL, R-LW, MN, OP, JR, ES, ASe, JPAS, PAu and SC) conceived and developed the concept for this scoping review collectively, as part of the Global Collective for Indigenous Adolescent Health and Evidence-Based Action. ASi, SE, SC, PAz, SW, JT-S, AB and RR developed the initial search protocol. PAz, AB, JT-S, SW, JK, DM, RR, BB, NB, CC-M, TCC, SE, SMF, KLH, MH, JMFK, SK, CL, R-LW, MN, OP, JR, ES, ASe, JPAS, PAu and SC reviewed the search protocol critically and provided input from their respective nations. ASi produced initial manuscript drafts, then ASi, PAz, AB, JT-S, SW, JK, DM, RR, BB, NB, CC-M, TCC, SE, SMF, KLH, MH, JMFK, SK, CL, R-LW, MN, OP, JR, ES, ASe, JPAS, PAu and SC undertook further drafting, critical review and editing to produce the final manuscript, and approved the final manuscript for publication. SC is responsible for the overall content as the guarantor.

**Funding** This work was supported by the Australian National Health and Medical Research Council (NHMRC) Centre for Research Excellence for Driving Investment in Global Adolescent Health (GNT 1171981). PAz is supported by an NHMRC Investigator grant (GNT 2008574). JPAS's participation was funded by FORTE: Swedish Research Council for Health, Working Life and Welfare (grant number 2021-01337). SK's participation was supported by funding from the Sámi National Centre for Mental Health and Substance Use (grant number S3-2020), and the Sámi Parliament of Norway (grant number RD006/20). JT-S and MH's participation was supported by the Medical Research Institute of New Zealand (grant numbers not applicable). TCC's participation was supported by a Cure Kids Professorial Chair (grant number not applicable). Funders have not been involved in the development of this protocol.

**Competing interests**  None declared.

**Patient and public involvement**  Patients and/or the public were not involved in the design, or conduct, or reporting, or dissemination plans of this research.

**Patient consent for publication**  Not applicable.

**Provenance and peer review**  Not commissioned; externally peer reviewed.

**ORCID iDs**
Andrew Sise http://orcid.org/0000-0003-0009-509X
Peter Azzopardi http://orcid.org/0000-0002-9280-6997
Jordan Tewhaiti-Smith http://orcid.org/0009-0001-5952-5181
Rachel Reilly http://orcid.org/0000-0003-2107-9187
Terryann C Clark http://orcid.org/0000-0001-5499-5080
Matire Harwood http://orcid.org/0000-0003-1240-5139
Odette Pearson http://orcid.org/0000-0001-9877-6509
Elizabeth Saewyc http://orcid.org/0000-0002-1625-9506
Jon Petter A Stoor http://orcid.org/0000-0002-1580-8307
Sue Crengle http://orcid.org/0000-0001-9367-1492

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
