## [Reviewer comments · BMJ Open]

ARTICLE DETAILS

TITLE (PROVISIONAL)	The health and wellbeing needs of Indigenous adolescents: A protocol for a scoping review of qualitative studies
AUTHORS	Sise, Andrew; Azzopardi, Peter S; Brown, Alex; Tewhaiti-Smith, Jordan Michael; Westhead, Seth; Kurji, Jaameeta; McDonough, Daniel; Reilly, Rachel; Bingham, Brittany; Brown, Ngiare; Cassidy-Mathews, Chenoa; Clark, Terryann; Elliott, Salenna; Finlay, Summer May; Hansen, Ketil Lenert; Harwood, Matire; Knapp, Jonil; Kvernmo, Siv; Lee, Crystal; Watts, Ricky-Lee; Nadeau, Melanie; Pearson, Odette; Reading, J; Saewyc, Elizabeth; Seljenes, Amalie; Stoor, Jon Petter; Aubrey, Paula; Crengle, Sue

VERSION 1 – REVIEW

REVIEWER	Grande, Antonio José Universidade Estadual do Mato Grosso do Sul
REVIEW RETURNED	15-Oct-2023

GENERAL COMMENTS	Dear Authors. The protocol is well written. 1- Have you done an initial consultation with a group of Indigenous to see if research questions make sense for them? 2- Change title to qualitative studies. 3- I believe a definition of study type is appreciate. For example: the way it is using mixed-methods, etc... lack a more rigorous definition. 4- Why not to assess risk of bias of included studies? 5- is there any Indigenous in the group of researchers? 6- Small detail to fix, MEDLINE (via PubMed)
---

REVIEWER	Okpalauwaekwe, U University of Saskatchewan, Medicine
REVIEW RETURNED	13-Nov-2023

GENERAL COMMENTS	It wa a pleasure reviewing your work. Overall, the protocol is well-structured, with a clear focus on an important public health issue. Enhancements on a few minor areas could further strengthen the quality of this work. See comments below. Specific comments Abstract The abstract provides a concise and coherent overview of the study, including its purpose, methods, and dissemination plans. However, while the abstract mentions the review's broad goals, detailing more specific objectives or research questions could enhance clarity. Also, a brief mention of the potential impact or
---

	significance of this review in the field could strengthen the abstract. Introduction Consider expanding on the existing gap in literature synthesis in this field to provide a more compelling argument for the study's necessity. Methods The scope of the review is appropriately wide, covering multiple countries and including a range of qualitative studies which is good. However, it would be good to provide more justification for the specific age range and countries included could add depth to the rationale behind the study design. Also, provide further elaboration on how the data will be synthesized and analyzed considering this wide scope. Conclusion The conclusion emphasizes the importance of the study and its potential impact on Indigenous adolescent health and appropriately connects the review's goals to broader efforts in improving health outcomes for Indigenous adolescents. Comments to consider: 1) Although it is a protocol, suggesting potential implications or future research directions based on the anticipated findings. 2) Addressing how the review will engage with Indigenous communities or incorporate Indigenous knowledge systems could enhance the study's relevance and impact. Remember there are rules for engaging Individual Indigenous communities. What are considerations to avoid a pan-Indigenous outcomes which could inherently be challenging to implement in communities where values, customs and culture differ.
--	---

REVIEWER	Beaulieu, Emilie Département de pédiatrie, Faculté de médecine, Centre Hospitalier Universitaire de Québec-Université Laval, Quebec City, Quebec, Canada, Pediatrics
REVIEW RETURNED	29-Nov-2023

GENERAL COMMENTS	Very interesting and important work. With your search terms, you are likely to come across a whole body of literature reporting on the health of Indigenous youth, but from an adult's perspective (qualitative methods). It would be very interesting to compare themes and results between adults' and teens' perspectives on their health. This would likely how different and why it is so important to let youth be heard.
---

VERSION 1 – AUTHOR RESPONSE

R1. 3	2- Change title to qualitative studies.	Thank you. The title has been changed to: ' The health and wellbeing needs of Indigenous adolescents: a protocol for a scoping review of qualitative studies '
R1. 4	3- I believe a definition of study	Thank you. We have added the underlined text on Page 7 to provide a more precise specification of the type of study that we are including:

	type is appreciate. For example: the way it is using mixed-methods, etc... lack a more rigorous definition.	We will include peer-reviewed and grey literature; in Australia for example there are a number of high quality consultations with Aboriginal and Torres Strait Islander adolescents that are published by Government and non-Government organisations.[23] To be eligible, studies must report data attained using qualitative methods (e.g.: interviews, focus groups, photovoice) where youth participants were able to identify and express their own views and perspectives. Quantitative studies including, for example, analyses of responses to surveys that used pre-set questions determined by researchers will not be included. Mixed methods studies (including both qualitative and quantitative components) will be eligible only if the health needs and/or priorities which are presented were identified based on qualitative enquiry with Indigenous adolescent people.
R1. 5	4- Why not to assess risk of bias of included studies?	Thank you. We seek to map the available evidence and literature in this area, rather than identify an answer to and quantify the level of certainty for a precise question (such as might be done in a systematic review). While we may discuss particular limitations of available literature (including publication bias) in our end publication, a formal assessment of methodological limitations, or risk of bias, will not be undertaken. This is consistent with recommended approaches for scoping reviews (See JBI manual for evidence synthesis, chapter 11.1.1: Why a scoping review? https://jbi-global-wiki.refined.site/space/MANUAL/4687794/11.1.1+Why+a+scoping+review%3F)
R1. 6	5- is there any Indigenous in the group of researchers?	Thank you. The research group is Indigenous led and includes indigenous researchers from across the included countries (see Lancet, 2021: https://doi.org/10.1016/S0140-6736(21)02719-7). As per our response to an earlier comment, we have added the following to the Background section (page 5): ...living within Australia, Aotearoa New Zealand, Alaska, Canada, the United States of America, Greenland, Norway, and Sweden. The Global Collective is led by Indigenous people, and includes Indigenous and non-Indigenous researchers and youth advocates from across these nations. And to the abstract (page 2): ...across Australia, Aotearoa New Zealand, Canada, the United States of America, Greenland, and Sami populations (Norway, and Sweden). A team of Indigenous and non-Indigenous researchers from these nations We will systematically search...
R1. 7	6- Small detail to fix, MEDLINE (via PubMed)	Thank you. We intend to search all databases accessed via the PubMed search engine. We have updated our wording regarding databases on page 8 of the manuscript as follows: A systematic electronic search will be performed of PubMed (including the MEDLINE, PubMed Central, and Bookshelf databases), CINAHL, Embase, Scopus, Informit (Indigenous and

		Health collections), Google Scholar, Arctic Health, the Circumpolar Health Bibliographic Database, Native Health Database (Canada and USA), iPortal (University of Saskatchewan), and NZresearch.org (Aotearoa New Zealand). Similar wording has been added to the relevant part of the abstract (page 2).
R2.1	It was a pleasure reviewing your work. Overall, the protocol is well-structured, with a clear focus on an important public health issue. Enhancements on a few minor areas could further strengthen the quality of this work. See comments below.	Thank you.
R2.2	Abstract The abstract provides a concise and coherent overview of the study, including its purpose, methods, and dissemination plans. However, while the abstract mentions the review's broad goals, detailing more specific objectives or research questions could enhance clarity. Also, a brief mention of the potential impact or significance of this review in the field could strengthen the abstract.	Thank you. Within available word limits, we have revised the abstract to be more explicit about the study's objectives (see tracked revisions in abstract, page 2). In particular, we highlight the following wording which has been added: This scoping review will seeks to understand the specific health needs and priorities of Indigenous adolescents aged 10 – 24 years captured map via qualitative studies conducted with Indigenous adolescents across Australia, Aotearoa New Zealand, Canada, the United States of America, Greenland, and Sami populations (Norway, and Sweden). We have also made the following changes to the Ethics and Dissemination section of the abstract, to briefly clarify that the review will help to inform broader collaboration to develop evidence-based solutions and actions for Indigenous adolescent health. Ethics approval was not required for this review. Findings will be disseminated via a peer reviewed journal article and will inform a broader international collaboration for Indigenous adolescent health project to develop evidence-based actions and solutions. to improve Indigenous adolescent health. Note that a number of other minor revisions have been made to the abstract to ensure clarity of wording and respond to other reviewers (see tracked revisions in abstract, page 2)
R2.3	Introduction Consider expanding on the existing gap in	Thank you for this comment, which has enabled us to make some important updates to our background section. This includes acknowledging a recently published systematic review (Int. J. Environ. Res.

	literature synthesis in this field to provide a more compelling argument for the study's necessity.	Public Health, 2022: https://doi.org/10.3390/ijerph192013688) within a similar subject area identified by a search undertaken during revision, and positioning our intended scoping review with respect to this. We have also explained the need to explore grey literature, and related this protocol to the wider workplan of the Global Collective for Indigenous Adolescent Health and Evidence Based Action. Edits to this effect have been made on Page 6: For example, qualitative studies have explored the health needs of Aboriginal and Torres Strait Islander adolescents living in rural settings in Australia,[15] perspectives of First Nations adolescent girls in Canada regarding the meaning of (and what is important for) a healthy body,[16] and Inuit youth perspectives on sexual health in Nunavut.[17] However, a search of the Pubmed database has revealed no recent synthesis of this rich qualitative data. Recently, substantial progress has been made in the synthesis of this evidence, with a systematic review of peer-reviewed literature having examined and compared key aspects of wellbeing for Indigenous young people aged 18 years or less in Canada, Australia, Aotearoa New Zealand and the United States of America.[18] To build on these efforts and set the foundation for the planned work of the Global Collective,[12] further exploration is necessary. In particular, it will be important to identify and map evidence across all nations included within the Global Collective's workplan, identify literature for youth across the Global Collective's broader age definition of adolescence, and explore the appreciable body of grey literature that may be available for some nations. Scoping reviews are well suited to exploring and identifying key concepts within a body of literature, as well as mapping, reporting and discussing these concepts.[19]
R2.4	Methods The scope of the review is appropriately wide, covering multiple countries and including a range of qualitative studies which is good. However, it would be good to provide more justification for the specific age range and countries included could add depth to the rationale behind the study design. Also, provide further elaboration	Thank you. We have included further justification for the age range and included countries, referencing our earlier concept paper which outlined the work which the Global Collective is doing. Regarding data synthesis and analysis, we have added further description to clarify that due to the wide scope, the analysis will have an exploratory element and will depend on the nature and type of data available. We have also further clarified that our synthesis of these studies will be descriptive, and will not utilize more complex methods of qualitative evidence synthesis such as meta-aggregation. We have made some minor adjustments to our wording regarding analyses to reflect this, and to help clarify that our identification and comparison of needs and priorities identified across studies will focus on the stated findings of those studies – i.e., identifying, grouping, and categorizing key findings to describe any overall needs and priorities which are apparent across identified studies, as well as identifying differences. The relevant changes are:

on how the data will be synthesized and analyzed considering this wide scope.	Background (Page 5, clarifying age range used for broader workplan of the Global Collective): Initially, the Global Collective will focus on Indigenous adolescents (defined for this purpose as being 10 – 24 years of age) living within Australia, Aotearoa New Zealand, Alaska, Canada, the United States of America, Greenland, Norway, and Sweden. Page 7 (changes to wording around study eligibility): As per United Nations practice,[2] a rigid definition of Indigenous status will not be used. Participants of a study will be considered Indigenous when this is clearly identified by the text of the associated article. While it is acknowledged that Indigenous peoples inhabit many regions of the world, the included regions/nations of Aotearoa New Zealand, Alaska, Australia, Canada, the United States of America, Greenland, Norway, and Sweden have been chosen for the initial work of the Global Collective due to similarities in their colonial histories, data systems, and policy contexts.[12] Currently, the World Health Organisation defines adolescence as occurring between the ages of 10 and 19 years.[22] However, evolving understandings of the timing of biological maturation, and increasing delays in role transitions such as marriage and parenthood mean that it is now proposed that adolescence should be defined as occurring between the ages of 10 and 24 years.[23] The Global Collective has chosen the age range of 10 – 24 years to reflect this understanding, and to capture the profound health-shaping changes which occur during this pivotal stage of development.[12] For this review, we will only include studies that specifically sample and report data for individuals ranging from 10 to 24 years of age. Page 10 (changes to wording around analyses): Analysis and presentation of results will be primarily descriptive, and efforts to assess the level of certainty of results or undertake more complex methods of synthesis such as meta-aggregation (as might be done in a qualitative systematic review) will not be undertaken.[20] Firstly, we will map the nature of the qualitative data overall for Indigenous adolescents. A descriptive table will be used to describe key characteristics of included studies including time of publication, age of participants, Indigenous peoples covered, and the focus of these qualitative studies. This will enable the available literature to be mapped and gaps in coverage to be identified. We will then undertake separate further analyses of the studies identified by this review. Due to the wide scope of included nations and data sources, these analyses will be inherently exploratory, and their final format will depend, depending on the quality and nature of data identified. For example However, we hope to be able to synthesize what the overall priorities and needs of Indigenous adolescents are from the descriptive data that have been identified
--	---

		from across the findings of our included studies. We plan to use a simple descriptive analysis of qualitative content[20] to synthesize the health and wellbeing needs and priorities which are identified across from the findings of included papers literature. We will aim to code needs and priorities identified by individual studies (for example, stress, worry, poor sexual health, racism), then group these into higher level categories (e.g., health outcome, health risk, health determinant), using a process similar to that undertaken by Harfield and colleagues.[29] Alongside describing any core health and wellbeing needs and priorities overall that are apparent across the findings of many studies, these categories will enable (where data are sufficient) simple comparisons across different demographic variables such as age group, country/region, and gender. A narrative description will be used to highlight the key findings of these analyses.
R2. 5	Conclusion The conclusion emphasizes the importance of the study and its potential impact on Indigenous adolescent health and appropriately connects the review's goals to broader efforts in improving health outcomes for Indigenous adolescents.	Thank you.
R2. 6	Comments to consider [conclusion, 1]: 1) Although it is a protocol, suggesting potential implications or future research directions based on the anticipated findings.	Thank you. While we do not know the implications at this stage, we have added some comment on potential implications in the review's conclusion (see changes on page 11): By identifying what is important to adolescents themselves, this review will highlight areas for action. In addition, we. It will also will map the coverage of literature in this crucial area, and establish whether gaps exist that warrant further research. Depending on the review's findings this could include, for example, identifying priority areas for consultation and primary qualitative enquiry, or set the direction for further evidence syntheses to develop evidence-based responses. This will be a key step in the wider efforts of our the Gglobal Ccollective,[12] which aims to measure, respond to, and develop strategies to enhance Indigenous adolescent health.
R2. 7	Comments to consider [conclusion, 2]: 2) Addressing how the review will	Thank you. In this review we hope to both identify overall priorities that are common to Indigenous adolescents in many settings, and compare priorities that exist across different contexts (e.g., age, country/region, gender) (see text on page 10, under heading 'Analysis and presentation of results'). A key strength of this review is

	engage with Indigenous communities or incorporate Indigenous knowledge systems could enhance the study's relevance and impact. Remember there are rules for engaging Individual Indigenous communities. What are considerations to avoid a pan-Indigenous outcomes which could inherently be challenging to implement in communities where values, customs and culture differ.	the large group of included researchers which includes experts from across included nations and regions. Work within each nation is being led by researchers and experts in that nation, enabling specific considerations for each nation to be addressed, as well as common themes to be identified across all nations which are included in the review. The following text has been added to the conclusion (page 11): Improving the health and wellbeing of Indigenous adolescents will be essential to addressing the inequities faced by Indigenous peoples.[9] With this scoping review, we will evaluate the peer reviewed qualitative literature pertaining to Indigenous adolescent perspectives on what is important for their health and wellbeing. Having begun with a preliminary search in October 2023, this review is progressing with work relevant to each nation being led by researchers and experts of those nations. This will enable consideration of issues specific to each region, whilst also allowing the identification of common themes and development of overarching recommendations. We have also added the following to the methods section (page 6 – within edits that also show the review timeline) to introduce this concept earlier within the protocol: The development of this protocol was informed by the Joanna Briggs Institute (JBI) Scoping Review Methodology Guidelines.[20] This scoping review will be reported in accordance with the Preferred Reporting Items for Systematic Reviews and Meta-Analyses extension for Scoping Reviews (PRISMA-ScR) checklist.[21] The scoping review which this protocol describes is underway. We ran the preliminary search in electronic databases in October 2023 and to date have completed a title and abstract screen based on eligibility criteria. Full texts are currently being retrieved for further assessment and we will also hand search grey literature. The review is being undertaken by researchers across the Global Collective, with work relevant to each nation being undertaken by researchers and experts of that nation. We anticipate the review will be completed in late 2024.
R3.1	Very interesting and important work. With your search terms, you are likely to come across a whole body of literature reporting on the health of Indigenous youth, but from an adult's perspective (qualitative	Thank you for this comment, which raises an important point. Our inclusion criteria restrict eligible studies to those in which youth were able to express their own views and perspectives; the findings regarding these views and perspectives will be the focus of this piece of work. While we agree that a comparison between youth and adult views is highly relevant, we anticipate that our exclusive focus on youth perspectives and views will mean that an effective comparison between youth and adult views will not be feasible as a part of this research. However, we acknowledge that even within studies that report youth views and perspectives, the perspective of adult researchers will influence study conduct and interpretation of findings. This is something that we will need to be mindful of at data extraction.

	methods). It would be very interesting to compare themes and results between adults' and teens' perspectives on their health. This would likely how different and why it is so important to let youth be heard.	Two relevant pieces of text have been added that help to address this issue: Page 7 (under heading 'Publication type'): We will include peer-reviewed and grey literature; in Australia for example there are a number of high quality consultations with Aboriginal and Torres Strait Islander adolescents that are published by Government and non-Government organisations.[24] To be eligible, studies must report data attained using qualitative methods (e.g.: interviews, focus groups, photovoice) where youth participants were able to identify and express their own views and perspectives. Quantitative studies including, for example, analyses of responses to surveys that used pre-set questions determined by researchers will not be included. Page 9 (under heading 'Data extraction'): Data extraction will be handled in two stages. First, for studies meeting inclusion criteria (Indigenous population AND adolescent focused AND qualitative in design) we will capture basic demographics of the study using a standardised form. This will include publication year, abbreviated reference, and whether the study focuses on understanding health and wellbeing, needs and priorities from the perspective of adolescents OR if the study focuses on a specific health issue/s (such as mental health). While this review is restricted to studies in which youth were able to identify and express their own views and perspectives, we will also consider the extent to which studies appear to reflect these perspectives and whether, for example, core findings appear based primarily on youth or the researcher's recommendations. In the second phase, we will first look at those studies that explore broad issues, needs and priorities, such as that conducted by Mohajer and colleagues.[15]
--	--	---

VERSION 2 – REVIEW

REVIEWER	Okpalauwaekwe, U University of Saskatchewan, Medicine
REVIEW RETURNED	13-Apr-2024
GENERAL COMMENTS	Thank you for the pleasure to review your revised work again. The quality has significantly improved and I don't have any major issues with this draft. I wish you the best and look forward to read your work in the future.